# Green Hydrogen Production through Ammonia Decomposition Using Non-Thermal Plasma

**DOI:** 10.3390/ijms241814397

**Published:** 2023-09-21

**Authors:** Julia Moszczyńska, Xinying Liu, Marek Wiśniewski

**Affiliations:** 1Department of Materials Chemistry, Adsorption and Catalysis, Faculty of Chemistry, Nicolaus Copernicus University in Torun, Gagarina 7, 87-100 Torun, Poland; jmoszczynska01@gmail.com; 2Institute for Catalysis and Energy Solutions, University of South Africa, Private Bag X6, Florida 1710, South Africa; liux@unisa.ac.za

**Keywords:** hydrogen storage, green hydrogen synthesis, non-thermal plasma, ammonia splitting, catalysis

## Abstract

Liquid hydrogen carriers will soon play a significant role in transporting energy. The key factors that are considered when assessing the applicability of ammonia cracking in large-scale projects are as follows: high energy density, easy storage and distribution, the simplicity of the overall process, and a low or zero-carbon footprint. Thermal systems used for recovering H_2_ from ammonia require a reaction unit and catalyst that operates at a high temperature (550–800 °C) for the complete conversion of ammonia, which has a negative effect on the economics of the process. A non-thermal plasma (NTP) solution is the answer to this problem. Ammonia becomes a reliable hydrogen carrier and, in combination with NTP, offers the high conversion of the dehydrogenation process at a relatively low temperature so that zero-carbon pure hydrogen can be transported over long distances. This paper provides a critical overview of ammonia decomposition systems that focus on non-thermal methods, especially under plasma conditions. The review shows that the process has various positive aspects and is an innovative process that has only been reported to a limited extent.

## 1. Introduction

### 1.1. Problems with Hydrocarbons

For centuries, the combustion of hydrocarbon fuels was the basis of processes used to obtain energy, but this has contributed significantly to the degradation of the environment. However, human ecological awareness has improved over the years, which is the main reason for the search for sources of renewable green energy that do not emit CO_2_ [1].

### 1.2. Ammonia as a Solution for Storing and Transporting Hydrogen

The demand for green energy has increased tremendously, and attention is increasingly being paid to using hydrogen as a fuel. This has both advantages and disadvantages. The main benefits are that hydrogen is non-toxic, has a high calorific value, is flammable in a wide temperature range, and is renewable [2] However, the ecological advantages are diminished by the process of transporting and storing the fuel, as the process is extremely difficult and expensive. Compressing hydrogen gas is energy- inefficient, while transporting liquefied H_2_ requires a very low temperature of 20 K [3]. The solution to this problem may be to obtain hydrogen through ammonia decomposition. Compared to compressed hydrogen, ammonia has a greater volumetric density, and the liquefaction process is easier. Therefore, this process may provide a more efficient method for storing hydrogen and so enable wide use in the energy industry [4].

Of course, it is possible to use organic compounds as an alternative; these include alcohols such as methanol, ethanol, propanol, butanol, and glycerol. However, they have a much lower power density; therefore, the process is less efficient than when using ammonia. The power referenced to hydrogen (100%) has shown maximum values as follows: 78% for butanol; 80% for glycerol; 83% for propanol; 86% for ethanol; and 93% for methanol. In the case of ammonia, the value is close to 100%. In addition, when organic compounds are used, the possibility of the formation of carbon by-products, which are often difficult to separate from hydrogen, should be considered, as hydrogen produced in this way is not green [5]. Cycloalkanes, especially cyclohexane and methylocyclohexane, are common hydrogen carriers. However, the high enthalpy required for dehydrogenation makes the hydrogen storage efficiency low. The same applies to the use of aromatic compounds. It has been proven that the introduction of a heteroatom into the ring lowers the enthalpy value while the efficiency increases. However, the issue of by-products still needs to be considered [6]. 

When obtaining hydrogen from methane, the most commonly used method is steam methane reforming, which generates a large carbon footprint [7]. Much attention is paid to light metal hydrides because they have a high-volume hydrogen density. However, large-scale application is impossible. Strong covalent and/or ionic bonds between metal and hydrogen atoms result in slow kinetics, poor reversibility, and a very high dehydrogenation temperature [8].

## 2. Hydrogen Fuel Cells

Fuel cells are devices that supply energy as a result of an electrochemical reaction. They do not emit carbon, nitrogen, or sulfur oxides, so they are suitable for use from an ecological point of view. The construction of the fuel cell is similar to that of a battery: it consists of two electrodes (an anode and a cathode) separated by an electrolyte. The fuel in the cell is hydrogen, which is transported (after dissociation) through the electrolyte to the cathode zone to form water [9].
(1)$$Anode\;process:\;H_{2}\to 2H^{+}+2e^{-}\;$$
(2)$$Cathode\;process:\;\frac{1}{2}O_{2}+2H^{+}+2e^{-}\to H_{2}O\;$$
(3)$$Total\;reaction:\;O_{2}+2H_{2}\to H_{2}O+energy\;$$

With these reactions, no environmentally harmful substances are formed during the process, enabling the generation of green energy, which is necessary in today’s world [10].

### 2.1. Negative Factors Associated with the Anode Process

The hydrogen fuel must meet certain purity conditions to achieve the highest possible efficiency. The presence of contaminants reduces the amount of energy produced and can lead to the degradation of the structural elements of the cell. Contaminants such as carbon oxides and organic and inorganic sulfur compounds can be produced during the reforming process [11].

The effect of CO and CO_2_ depends on the type of catalyst used. It was shown that these oxides bind well with metal surfaces, which caused a decrease in the number of active centers available for hydrogen. After adsorption, carbon dioxide can be catalytically reduced to carbon monoxide, which poisons the catalyst. This effect depends on many factors, including temperature, pressure, concentration, exposure time, type of anode catalyst, and current density [11,12,13]. In the case of CO poisoning, introducing ruthenium to the catalyst increases its tolerance to poisoning and thus eliminates the harmful effects. Hydrogen sulphide causes an even greater yield loss, which has a stronger poisonous effect on noble metal catalysts. Even admixtures of other metals, such as ruthenium, do not reduce this effect [11].

The essential air contaminants are nitrogen oxides (NO_x_). Even small amounts cause a decrease in the efficiency of the cell, which further deteriorates as the concentration of nitrogen compounds increases. Long-term exposure to NO_x_ may cause the cell to malfunction [14]. Other air pollutants in the cell are sulphur oxides, which increase the acidity and thus change the electrode potential [12]. 

Organic compounds also have a negative effect on fuel cell efficiency. Exposure to benzene causes a temporary drop in performance, but this is fully recovered after the cell is cleaned. Toluene also reduces the service life of the devices. Contrarily, specific contaminants improve the performance of the cells. One example is ozone, which is a strong oxidant that generates improved performance, especially with a low current density. Moreover, the presence of ozone in the air can eliminate disruptive organic compounds [11]. 

It should be remembered that there is rarely a single pollutant at play; the effect is really the result of all the pollutants acting on the cell [11,14]. Even a superficial review shows that many different factors affect the workings of a hydrogen fuel cell. Most of them are negative, but there are ways to eliminate or minimize them. The presence of pollutants may be limited by choosing a hydrogen production method with by-products that do not contain impurities. One good choice is ammonia decomposition, which allows CO_x_-free hydrogen to be obtained. 

### 2.2. Colors of Hydrogen

Hydrogen, especially green hydrogen, is an essential element in a fuel cell. It plays a key role and can be differentiated from the others, i.e., purple, black, and turquoise. However, the main types are grey, blue, and green (Table 1). Grey hydrogen is produced by hydrocarbon reforming and has a high CO_x_ emission. Blue hydrogen also comes from fossil fuel processing, but the process is enhanced using emission-reducing CO capture methods. Green hydrogen is produced during electrolysis and has zero emissions [15,16].

Much research on obtaining hydrogen by electrolysis of water/seawater has been reported [17,18,19,20]. It suggests that this is a promising method, but the storage and transportation problems still exist. Therefore, the preferred method of hydrogen production may be via decomposition of ammonia. Hydrogen could be transported in ammonia and the reforming process could occur at the destination.

There are many different ways to obtain hydrogen from ammonia. This paper provides a critical overview of the advances made in ammonia decomposition, with an emphasis on plasma processes.

## 3. Ammonia Decomposition Mechanisms

### 3.1. Possible Pathways of the Ammonia Decomposition Mechanism

There are many options for using catalysts in ammonia decomposition by using plasma. The efficiency and kinetics depend on the catalyst type used, while the mechanism is similar in most cases. Initially, the ammonia adsorbs to the surface of the catalyst. Next, the bonds between the hydrogen and nitrogen atoms are gradually activated. As a result of their breaking, free N and H atoms are formed. These adsorb on the catalyst surface and then combine to form N_2_ and H_2_ molecules, which desorb from the catalyst surface. The mechanism for obtaining hydrogen by ammonia decomposition can be represented by the following elementary reactions [21,22]:(4)$$NH_{3}{+}^{*}⇌NH_{3}*$$
(5)$$NH_{3}*{+}^{*}⇌NH_{2}*+H*$$
(6)$$NH_{2}*{+}^{*}⇌NH*+H*$$
(7)$$NH*{+}^{*}⇌\;N*+H*$$
(8)$$H*+H*⇌H_{2}↑+2*$$
(9)$$N*+N*⇌N_{2}↑+2*$$

The path shown in Figure 1 indicates the simplest model of the ammonia decomposition reaction. However, experimental research has shown that the process is more complex, and many investigations have shown that, under specific conditions, it is possible to form hydrazine as a by-product of the reaction via a competitive reaction pathway [22]. Dirtu et al. [23] studied the mechanism of ammonia decomposition but concentrated on the formation of hydrazine and characterized the entire process using far-infrared and UV-Vis spectroscopy. They noted that protons and radicals could react with each other at different stages, which results in a process that is highly complicated. Further research conducted by Humblot et al. [24] shows that the direct conversion of ammonia to hydrazine is promising; however, this is a difficult process to carry out because the cleavage of hydrazine is thermodynamically more feasible than the degradation of the N-H bond in NH_3_. Therefore, it can be concluded that any catalyst that is capable of activating NH_3_ will also decompose N_2_H_4_.

It has also been shown that dissociated hydrogen does not necessarily recombine with another proton; it can react with NH_2_ to regenerate ammonia. Hanes et al. [26] focused their research on this step. They observed that, in the first stage of the reaction, dissociated protons and NH_2_ radicals were formed, but the radicals disappeared in the second step. They noted that NH_2_ does not react with NH_3_ because this kind of reaction causes a significant energy loss. No NH radicals were noticed in the reactor, so the only possibilities were the recombination of ammonia with NH_2_ and H or the formation of hydrazine from two NH_2_ radicals.

### 3.2. Mechanism of Ammonia Decomposition on Noble Metals

The most popular catalyst among noble metals is ruthenium. Consequently, many studies have focused on the mechanism and kinetics of ammonia decomposition on its surface. The reaction kinetics are dependent on the conditions of operation, but recombinant nitrogen desorption has been shown to be the rate-determining step in the decomposition of ammonia in all cases [27,28]. The activity of ruthenium as a catalyst depends on the support used, which may include carbon nanomaterials like nanotubes or metal oxides. Yin et al. showed that using carbon nanotubes delivers the best results [29]. Zhou at. al. investigated the mechanism of ammonia decomposition with Ru anchored on carbon nanotubes being used as the catalyst (Figure 2) [30]. The research focused on initial ammonia dehydrogenation and N_2_ recombination. Other noble metals that can be used as a catalyst for ammonia decomposition are platinum, palladium, iridium, and rhodium [29].

The catalytic decomposition of ammonia on Ir was investigated by Xiao et al. [31], with a focus on the mechanism and kinetics of the process. The results showed that, as in the case of ruthenium, nitrogen recombination is the stage that determines the rate of the reaction. They proved that the dominant surface form in this reaction is –NH_2_ for two reasons: the difficulty of obtaining the third dehydrogenation product (–NH) and the difficulty in the recombination of the NH_2_ radicals.

### 3.3. Mechanism of Ammonia Decomposition on Non-Noble Metals

Iron, nickel, and cobalt are cheaper catalysts that produce good hydrogen production results. Lanzani et al. [21] investigated the mechanism of obtaining hydrogen from ammonia on an iron catalyst. The first stage is the adsorption of ammonia on the iron surface, with three adsorption sites available: the top, hollow, and bridge sites. Ammonia is adsorbed most strongly at the top. This is followed by the dehydrogenation of ammonia, resulting in the formation of NH_2_ and H, which are co-absorbed in the hollow and bridge sites. The next step is the second round of dehydrogenation and the formation of NH and H. The investigated dehydrogenation pathways are exothermic, and the first round of dissociation is the step that determines the rate of the reaction. This is because the Fe-N interaction does not weaken the N-H bond significantly.

Duan et al. [32] compared the mechanisms of action of three transition metals as catalysts in the process of ammonia decomposition, i.e., iron, nickel, and cobalt, in the form of close-packed surfaces. Their research was aimed at selecting the most effective non-precious metal catalyst. The results showed that cobalt and nickel could deliver a higher ammonia decomposition rate than iron because of having higher d-band centers (Figure 3).

The stronger the metal–nitrogen bond is, the easier it is for ammonia dehydrogenation to take place, but the more difficult it is for nitrogen recombination to happen. In the case of iron, the high barrier to recombination causes nitrides to form on the catalyst surface. This has led to research on the possibility of using metal nitrides as catalysts in the ammonia decomposition process, and it has been found that nitrides often perform better than pure metals [33].

Zhao et al. [34] investigated the mechanism of ammonia decomposition over molybdenum nitrides and reported that during the first stage of dehydrogenation, NH_3_ dissociates to NH_2_ and H. The first activation of the bond between hydrogen and nitrogen can occur in any of three pathways, depending on the adsorption site. The second stage of dehydration leads to another hydrogen atom and NH forming, and the third to the creation of N and H adatoms, with N- remaining adsorbed on the catalyst. Hydrogen adatoms recombine and form the H_2_ molecule, which can easily be desorbed from the surface (Figure 4).

There are two types of nitrogen atoms on the surface of molybdenum nitride: native and adsorbed nitrogen. As the native nitrogen is involved in the overall process, there are a few possible combinations for the desorption of N_2_ molecules. Both atoms may derivate from ammonia dissociation—this is called the Langmuir–Hinshelwood mechanism. Both atoms may be from native surfaces, or the atom that comes from the dissociation can be combined with the native-surface N atom, according to the Mars–van Krevelen scheme.

The literature does not report on plasma-assisted combustion, and only a few papers refer to ammonia decomposition being achieved in this way. Aside from obtaining hydrogen, the first experiments involving the use of plasma were also aimed at generating hydrazine [35]. It is possible to use non-noble bimetallic oxide nanocatalysts, such as FeCr, that show high catalytic activity. Moreover, ammonia conversion was nearly 100%. The dual metal catalysts clearly outperformed the single metal samples in terms of their catalytic performance [36].

### 3.4. Mechanism of Ammonia Decomposition on Carbonaceous Materials

Recently, it was shown that pure carbonaceous materials (i.e., without the metal active phase) can be harnessed effectively for use as catalysts in hydrogen production. The data indicated that the tested materials possess excellent catalytic ability for economical, CO_x_-free hydrogen production from NH_3_ decomposition at a low temperature [37]. The authors stated that the reaction occurs thanks to the innate H resistivity and N affinity of the carbon materials. These key factors shift the equilibrium to the product, which allows almost 100% conversion. Moreover, the process is performed only at the surface and does not interfere with the C-structure, which means the C-catalyst used is highly stable.

## 4. Thermal Methods Used in the Decomposition of Ammonia

### 4.1. Non-Catalytic Conversion

Thermal treatment is the most commonly used method for decomposing ammonia. The thermal decomposition of ammonia to hydrogen and nitrogen is an inverse reaction of the Haber–Bosch synthesis [25]. It is an endothermic reaction with an enthalpy of 92.4 kJ/mol [38].

It is possible to split ammonia thermally without using a catalyst, but the reaction temperature is higher than 677 °C [39]. This is the main reason for using a catalyst that allows for a lower decomposition temperature, as using a high temperature requires both large amounts of consumables and energy as well as specialized and expensive equipment. Introducing an inexpensive catalyst into the system significantly reduces the cost of obtaining hydrogen.

### 4.2. Catalytic Conversion

Ruthenium was the first material used as a catalyst [25], and it is still the most commonly used option. However, it may not be the best option. While it has high catalytic activity for the decomposition of ammonia, a question that needs to be asked is: does its use reduce the cost of obtaining hydrogen? Although ruthenium is one of the cheaper precious metals, its cost is relatively high when compared to other catalysts. Therefore, the process is not profitable enough to be used on a mass scale. This has led to a search for cheaper catalysts that do not reduce efficiency [40].

The possibility of using metals such as iron, nickel, molybdenum, cobalt, or magnesium has been tested using the pure metals and their mixtures. The active phase has also been embedded on different supports in the research literature [41,42,43]. Bell et al. [41] developed a cobalt catalyst deposited on γ-Al_2_O_3_. The relationship between cobalt particle size and catalytic activity was analyzed.

Iron is a viral catalyst that is also used in Haber–Bosch synthesis, which leads to the question: why not use it in the reverse reaction? Xun et al. [42] investigated the effectiveness of using iron in ammonia decomposition, using cobalt as a catalyst. Both of these metals tend to form agglomerates, so they need a promoter to act as an effective catalyst. Xun et al. [42] used lanthanum, and the research showed that Fe and Co had good activity and stability.

Pinzón et al. [43] focused their research on using Ni- and Co-based perovskites as catalysts and showed that bimetallic perovskites performed very well. The Co content influenced the catalytic properties, which improved the conversion of ammonia to hydrogen. Additionally, the samples were doped with Mg and Ce, which also affected hydrogen production efficiency positively.

Another method of reducing the decomposition temperature, and therefore the cost, is that of using membrane reactors. The process involves decomposing ammonia into hydrogen and nitrogen and then separating them. Using a membrane reactor allows these two steps to be carried out in one device. Cechetto et al. [44] adopted this technology using a Pd-membrane reactor with Ru as the catalyst. The results recorded when using the membrane reactor were compared to the results obtained when using a conventional reactor. The effect of temperature, permeate pressure, and flow rate on the efficiency of the processes was analyzed. It was found that introducing the membrane into the reactor increased its efficiency and enabled the complete conversion of ammonia above a temperature of 425 °C. The application of a vacuum accelerates the kinetics of the process.

The efficiency of hydrogen production and the purity are unquestionable advantages of this method; however, the issue of temperature and pressure must still be considered. Admittedly, at temperatures above 425 °C, the conversion of ammonia is practically complete, even at atmospheric pressure; but below 400 °C, a vacuum is necessary. Studies show that increasing the temperature increases the yield and purity of the obtained hydrogen. Using a Pd-based membrane limits the ability to accurately analyze the effect of temperature on ammonia decomposition because Pd decomposes above 500 °C. These factors increase the production cost, mostly via the use of expensive catalysts. In summary, the application of the membrane reactor produces satisfactory results, but a much cheaper method is needed to effectively reduce the combustion of fossil fuels that does not require the use of complex equipment.

Cechetto et al. [45] tried to reduce the cost of the process by increasing the thickness of the membrane and using a small purification unit in the permeate of the membrane. The main goal was to increase the purity of the hydrogen because membrane reactors do not always deliver a product that is pure enough for use in fuel cells. The results showed that a thicker membrane caused a decrease in the efficiency of the process, with a simultaneous increase in the purity of the obtained hydrogen.

Microwaves can also be used in a heating method, and this requires less energy than traditional heating methods. The most important advantage when using a microwave reactor is greater control of the process, as it is easy to switch a microwave on and off. Ammonia decomposition is possible in a microwave reactor at a temperature of 390–490 °C. Depending on the catalyst used, this reduces the cost of hydrogen production.

Many types of catalysts can be used in this process, and Varisli et al. [46] used iron incorporated into mesoporous carbon in their experiment and compared the results obtained using a conventional reactor and a microwave reactor. Seyfeli et al. [47] used carbon-supported cobalt in microwave ammonia decomposition, and another study was conducted using a nickel-based catalyst [48]. However, a microwave reactor has certain disadvantages, as not all materials can be heated by microwaves, which somewhat limits the choice of catalyst.

## 5. Non-Thermal Plasma Applications in NH_3_ Decomposition

Plasma is the fourth state of matter, comprising ionized particles in the form of neutral or charged elementary particles (Figure 5). Plasma shows a wide variety of quantitative and qualitative characteristics compared to other states of matter. However, all sets of mobile particles cannot be termed plasma because several conditions must be met. For instance, some conditions include the following: the particles must be able to exist in an excited state; the particles must meet the condition of electric quasi-neutrality and must have an appropriate concentration of electric charge carriers; and kinetic energy within a specific range is required [49].

Plasma stands out from other states; here, the physical and chemical interactions are intertwined, and very often, a change in its chemical composition causes a change in its properties. When there are only electrically charged particles in plasma, it is in a pure state; when neutral particles are introduced, it is in a mixed state. In general, even the most straightforward systems consist of a ‘great zoo’ of different chemical and physical entities.

There are many different types of plasma, but the primary categorization is of thermal plasma and non-thermal plasma. Another consideration is the power supply, which is an integral part of any plasma source. The most important groups include a high-voltage DC power supply and a low- and high-frequency AC power supply [49].

This review focuses on the low-temperature plasma process as a non-thermal method of decomposition of ammonia to hydrogen and nitrogen. There are miscellaneous types of plasma reactors, which generally do not have complicated structures; nor do they require high energy expenditure. This makes them more economical than a high-temperature system, which increases the cost of obtaining hydrogen. One advantage that distinguishes this method is the satisfactory control over the kinetics of each known elementary step in the process.

Plasma is made from different types of entities, and the temperature of each component can be described separately. Electron energy is estimated at 40,000 K [50], while the rest of the particles stay slightly above room temperature [51]. Thus, the temperature of the entire system is low enough that it remains non-thermal.

Another advantage of a plasma system is that you can easily turn the processes on and off. The overall temperature is low enough even for the adsorption of ammonia onto the porous catalyst. This increases the amount of substrate available and shifts the equilibrium to the product side, which raises the efficiency of the cracking process.

When a catalyst process is not used to decompose ammonia, the plasma yield is small and unsatisfactory. However, introducing a catalyst into the system significantly increases the yield. In addition, when selecting appropriate parameters, this method is characterized by high selectivity in hydrogen production. Thus, selecting an appropriate catalyst allows for obtaining hydrogen on a large scale more economically. It is worth noting that the catalyst that shows the best activity will not necessarily be the most suitable since the most active catalyst—ruthenium—is a very expensive choice [25].

This field of study is showing dynamic development and delivering promising results. Various types of catalyst plasma reactors are being tested in decomposing ammonia, with pure hydrogen being obtained. However, only a small number of studies are being conducted, which leaves the field wide open to exploration.

The most non-thermal plasma generating systems use an electric field as an excitation source. The next part of this paper discusses the basic types of plasma reactors, including their properties and applications.

## 6. Types of Plasma Reactors

Recently, the discharge chemistry and physics fields have been the most widely studied. Generally, it is usually a non-equilibrium process, and the nature of the discharge depends on the construction of the specific reactor. It is possible for surface, volume, or surface–volume discharge to occur [52].

### 6.1. Thermal Plasma

Thermal plasma is created by accelerating electrons between two electrodes in a gaseous environment. It is characterized by high electron density and a high temperature—usually in the range 10^6^ to 10^7^ K—with the value being dependent on the degree of gas ionization [53].

Thermal plasma causes a local thermal equilibrium to occur. Collisions between electrons and molecules transfer kinetic energy and, consequently, are the reason for the temperature of the environment. Assuming a given electric input, equilibrium is achieved when gas particles reach an energy level equal to the energy of the electrons [54,55].

The use of thermal plasma in technology is a topic that is frequently discussed by researchers in the field, given the importance of energy efficiency and the efficiency of processes with reduced environmental impact. However, a literature review showed that the use of thermal plasma is not widely used as a method for producing pure hydrogen by decomposing ammonia.

### 6.2. Glow Discharge

Glow discharge occurs when electric current flows through a gas under reduced pressure, which often occurs in a glass tube. The simplest type of discharge occurs when a direct current is applied. Initially, a small number of atoms in a reactor are ionized; then the positively charged ions move toward the cathode while the electrons move to the anode. Collision with other atoms causes ionization or excitation. Electrons and neutral atoms collide with the cathode and transfer some of their kinetic energy to it. This causes electrons to be ejected from the cathode, which is then accelerated to the glow discharge mass. The excited atoms lose energy by emitting a photon. Bright areas occur in the reactor, where light emission and dark spaces occur [56,57,58]. Glow plasma is a typical NTP because the gas temperature is much lower than the vibration temperature [59]. An additional advantage is that when using low gas pressure, glow discharge is an inexpensive method [60].

Glow discharge has been used in many different branches of science, such as in plasma polymerization [61] and carbon dioxide splitting [59]. The literature review has revealed that it is possible to obtain hydrogen by glow discharge plasma [62]. However, none of the reviewed papers focused on ammonia decomposition, with the majority preferring electrolysis with a glow discharge electrode. Typically, organic compounds are subjected to this type of process.

Yan et al. [63] obtained hydrogen by means of glow discharge plasma electrolysis. With this process, electrolysis occurs at the plasma/electrolyte border. Using plasma in the electrolytic process resulted in an increase in electrode efficiency compared to the conventional electrolysis process. The products obtained in the reaction were hydrogen, carbon monoxide, methane, ethane, propane, formaldehyde, and water. The authors proved that using glow discharge plasma yields a higher concentration of hydrogen with less energy consumption and fewer CO_2_ emissions than when using a dielectric barrier discharge or corona discharge plasma.

A few papers were found that report on the possibility of producing hydrogen using plasma glow reactors. Therefore, it is worth paying attention to this topic, as it may provide a tool that can be used for the large-scale decomposition of ammonia.

### 6.3. Corona Discharge

Corona discharge is also referred to as a glow discharge. It occurs at a pressure close to atmospheric pressure, with large distances between electrons in a system with large asymmetry of the electric field. The reactor consists of a high-voltage electrode with the source of the discharge—the corona—around it [63]. The corona discharge can be positively or negatively charged, depending on the electrode’s polarity. In the case of a positive corona, the charge carriers are ions, while the negative carriers are electrons. The differences between these two phenomena result in the differences in the masses of the charge carriers.

The discharge formation mechanisms also differ slightly [64]. The positive discharge is evenly distributed around the electrode surface, and free high-energy electrons are concentrated close to its surface. It propagates at a lower voltage value than the negative discharge and is divided into two regions: the plasma region around the electrode, where the electrons are stored, and the region farther away from the electrode, where the heavier positively charged ions are located.

A negative corona is unevenly distributed and is usually more significant than a positive corona. A characteristic feature of a negative discharge is the formation of secondary electrons due to successive collisions. The electrons do not concentrate around the electrode, and their energy is lower than in the case of a positive discharge. The negative discharge area is divided into three regions: an inner site closest to the electrode that contains ionizable high-energy electrons; an intermediate area with plasma that does not cause further ionization; and an outer space that contains negative ions. The main difference between the mechanisms of positive and negative discharge is the source of electrons in the corona: it comes from the corona electrode in the case of a negative discharge and from collisions in the area outside the plasma in the case of a positive discharge [64].

Corona discharge has been studied by numerous researchers [65], but it has never been used to obtain hydrogen through ammonia decomposition in plasma. However, Zou et al. [66] proved that it is possible to obtain hydrogen using corona discharge via dimethyl ether (DME) decomposition. The researchers investigated the flow rate, frequency, and waveform effect on the decomposition process. They concluded that adding neutral gas is the key factor to increasing the production rate. The observed effect is due to the promotion effect in the charge transfer. When pure DME is introduced into the reactor, the molecules are excited directly by the electrons. Adding an inert gas changes the reaction path: the atoms of the added gas are excited and then collide with the DME molecules, causing them to be excited in turn. Studies show that diluting the DME with an inert gas improves the decomposition efficiency.

The efficiency of corona reactors can be increased by combining their operation with the use of a catalyst, and catalyst selection depends on the process used. An example is the toluene cracking reaction, where a composite MnO_2_/CeO_2_ catalyst has proven to be an excellent example [67].

Although corona discharge is not yet a proven method for decomposing ammonia, the potential use of this reactor for producing hydrogen should be considered given the positive results seen when decomposing other substances. An example is in the decomposition of organic wastewater [68], where the process is of significant importance in terms of the environment as it can increase the biodegradability and degradation of organic pollutions including carcinogenic perfluorosurfactants (PFS) and other water contamination elements [69].

These examples suggest that corona discharge is a promising method for ammonia decomposition. Research in this direction would ensure development in the field of hydrogen production. The process has been used in many investigations that focused on producing hydrogen from organic compounds; therefore, the possibility of obtaining it from ammonia via this method should be investigated.

The main disadvantages of using the corona discharge method are corrosion, degradation, and fatigue behavior of the electrode, which can affect the performance of the processes negatively. Cádiz et al. [70] studied the degradation of corona electrodes. They showed that the surface of the electrode is destroyed after a short time of use regardless of the material from which it is made. The best, but not ideal, option is a tungsten electrode that allows the life of the electrode to be extended, which is crucial for the use of this option in industrial applications.

### 6.4. Dielectric Barrier Discharge

The discharge occurs between two electrodes, at least one of which must be covered with a dielectric material that constitutes a barrier. Typically, glass, ceramics, quartz, mica, or alumina are used for this. Plasma is formed in a tubular cylindrical reactor, and a voltage that varies over time is applied to the electrodes (Figure 6). By covering the electrodes with a dielectric material or placing a barrier between them, the discharge is evenly distributed over the entire surface of each electrode. Furthermore, the dielectric barrier eliminates sparks and prevents electrode degradation by etching or corrosion [71,72].

The current intensity depends on the carrier gas that is introduced into the reactor, and other important factors are the distance between the electrodes and the pressure in the reactor. The dielectric barrier discharge (DBD) method can be used at atmospheric pressure, but requires a shorter distance between the electrodes. A reduction in pressure makes it possible to increase this distance and, consequently, the length and volume of the reactor [73,74].

Much research has been conducted on the degradation of air or water pollution by using plasma. However, the literature does not report on plasma-assisted decomposition, and only a few papers refer to ammonia decomposition being achieved this way.

El-Shafie et al. [75] explored the possibility of using alumina particles as catalysts in the ammonia decomposition process using DBD plasma. They investigated how the size of the alumina particles affects the amount of hydrogen obtained. The hydrogen concentration and conversion rate were also assessed, as were the effects of the ammonia feed flow rate and the applied plasma voltage. The experiment was carried out under atmospheric pressure using a gaseous mixture of ammonia and argon. The concentration of hydrogen and nitrogen was controlled by using gas chromatography. The results showed that smaller catalyst particles work better in ammonia decomposition, as a smaller size entails a larger area. Thus, a larger amount of ammonia adsorbs on the catalyst and shifts the reaction equilibrium to the product side.

In another study, El-Shafie et al. [76] investigated the energy and exergy of hydrogen production by using non-thermal plasma methods including DBD. Three ammonia decomposition systems (PMR, CR-PMR, and CR-CPMR) were observed at different ammonia input flow rates. It was found that both energy efficiency and exergy were lowest with PMR and highest with CR-CPMR.

Lu et al. [33] investigated the effect of double-DBD plasma on the hidden active phase. They compared thermal catalytic ammonia decomposition and decomposition in plasma.

Andersen et al. [77] also used DBD plasma to obtain hydrogen via ammonia decomposition using a coaxial packed-bed DBD reactor. Both electrodes were made of stainless steel. The inner electrode was a rod while the outer electrode was a mesh wrapped around the reactor. The dielectric barrier was made of quartz and the packing barrier of glass wool. The experiment was conducted under atmospheric pressure without additional heating. Different catalytic materials were used, and it was concluded that MgAl_2_O_4_ is more active than TiO_2,_ SiO_2_, γ-Al_2_O_3,_ or BaTiO_3_. The effect of catalyst type and particle size on ammonia conversion was then analyzed, and it was concluded that the impact was minimal. However, as the particle size increases, the process of ammonia conversion decreases slightly.

Wang et al. [78] proposed using cheap metals such as Fe, Co, Ni, and Cu as a catalyst to improve conversion efficiency significantly. The key factor with these metals is the possibility of nitrides forming. The process mechanism shows that the recombination of adsorbed nitrogen is the slowest stage. The catalytic effect of the metals used depends on the strength of the metal–nitrogen bond, and the authors concluded that it decreases in the following order: Fe–N > Co–N > Ni–N > Cu–N.

Akiyama et al. [79] tested different materials (such as C, Cu, Ag, Al, Fe, and Ti) applied as inner electrodes and reported the similar activity of the tested catalysts. The reason may have been that when the reaction occurs only in the gas phase, regardless of the presence of potential catalysts (the tested materials are not the catalysts), the reaction is fast enough to produce a new, rich equilibrium, irrespective of the catalyst surface. Nevertheless, the authors did not perform zero-control analysis, and thus, both aspects of the reasoning are possible. The authors concluded that residue time and plasma power are the most important factors affecting the H_2_ yield.

A high degree of conversion of ammonia to hydrogen (82%) was achieved by Andersen et al. [80]. The kinetics of the process and the factors influencing it were also investigated, with the effects of changing the flow rate, power, and discharge volume being tested.

However, it is not just metal catalysts that can be used for ammonia decomposition by using plasma. Wang et al. [81] applied Mo_2_N and obtained very pure CO_x_-free hydrogen for fuel cell applications. The synergistic effect of using a catalyst in combination with DBD plasma resulted in near-complete NH_3_ conversion. The size of the Mo_2_N particles was not insignificant, as the powder catalyst showed the highest reactivity [81].

### 6.5. Gliding Discharge

Gliding arc discharge is characterized by high electron density, power, working pressure, and electron temperature. It is a non-equilibrium process; thus, it does not require vacuum to produce reactive forms already under atmospheric pressure. The plasma produced via gliding is weakly ionized. The high energy of the electrons in relation to the energy of the ions causes a local thermodynamic imbalance. Gliding allows thermal and non-thermal plasma areas to be generated; however, due to the properties of the non-equilibrium areas, the use of gliding plasma is classified as a non-thermal method [82].

Gliding discharge reactors are made of two electrodes, separated by a gap, between which an electric arc is generated. The transverse gas flow causes arc elongation and a higher power regime. Furthermore, gliding allows more power to be used, so it is considered a better method than corona discharge, especially for oxidation processes.

A gliding arc discharge can be used for hydrogen production in many different ways, but only one paper refers to ammonia decomposition. Lu et al. [83] used a combination of traditional gliding and supersonic/subsonic discharge (Figure 7) named the Laval nozzle arc discharge (LNAD). They investigated the effect that the applied voltage, the input gas flow rate, and the inlet methane and water vapor concentration had on the process. The same studies were repeated for methane reforming as a method of obtaining hydrogen.

### 6.6. Hybrid Reactors

The plasma membrane reactor is actually a type of hybrid reactor. It is created by introducing a selective membrane into a plasma device using various types of power supply. The membrane increases the efficiency of the plasma process because it limits the regeneration of ammonia from decomposition products. The rate of hydrogen formation is directly proportional to the value of the applied voltage. However, this is determined by the peak value at which nitrogen and hydrogen react to form ammonia. The membrane prevents this by removing the produced hydrogen from the system [84].

In the case of hydrogen, palladium-based membranes are used most often. There are many options (e.g., Pd-Cu, Pd-Ag, and Pd-Au) that can be used to construct an effective membrane, which is usually placed coaxially in the center of a quartz tube, which is one element of a reactor. There must be a gap between the reactor wall and the membrane because that is where the plasma is generated [85].

The mechanism of hydrogen separation requires it to be broken down into protons, which involves several stages. First, the NH_3_ molecules adsorb on the membrane surface, and then, they dissociate into atoms. Thus, the H atoms dissolve in bulk and diffuse onto the permeable side of the membrane, where they recombine into H_2_, and finally desorb from the membrane surface [84,85].

Hayakawa et al. [84] obtained hydrogen using a hybrid membrane–plasma reactor. The membrane was an alloy of Pd and Cu, and the palladium content was 40%. The plasma was generated by means of dielectric barrier discharge. A hydrogen permeability value of 80% was achieved. The method used made it possible to obtain hydrogen without the use of high temperature and a catalyst.

In another study [86], the authors used a combination of a plasma membrane reactor and a catalytic decomposition reactor. They applied the same Pd/Cu membrane as in a catalytic tube reactor, with Ni@Al_2_O_3_ being used as the catalyst (Figure 8). Most of the ammonia stream had already been decomposed in a catalytic reactor. (In contrast, the ammonia residue can be decomposed in the plasma). This method produced hydrogen with a very high purity value of 99.99%.

## 7. Summary and Conclusions

The literature review revealed the possibility, mechanisms, kinetics, and methods of ammonia decomposition that are used to produce pure hydrogen. In our opinion, ammonia is an efficient method of storing hydrogen because its production and decomposition require less energy than liquefying hydrogen.

The decomposition process mechanism was analyzed and a simple reaction path for the decomposition of ammonia was revealed. In the simplest terms, it comprises the adsorption of ammonia on the catalyst surface, gradual dehydrogenation, and the recombination of H_2_ and N_2_, which desorb from the surface. However, it has also been shown that decomposition could occur via the formation of intermediates such as hydrazine. In addition, dissociated hydrogen can also react with radicals at various stages of the process. Thus, by harnessing catalysts with low N affinity, like carbons [39], it is possible to raise the selectivity up to 100%.

This research focused on the reaction kinetics and, therefore, does not allow for an unambiguous definition of the stage that determines its speed. This is highly dependent on the catalyst used; however, in most cases, nitrogen recombination is the slowest step. Nevertheless, each catalyst interacts differently with ammonia molecules, and there are reports of the rate-determining step being the first dehydrogenation.

The review shows the broad range of methods that can be used for ammonia decomposition. It has been shown that thermal processes require high energy and material consumption and hence are not ecologically friendly or economical. Particular attention was paid to non-thermal methods, chiefly plasma methods. The results suggest that plasma holds promise for obtaining green energy. However, the topic is relatively new; hence, the literature on this topic is still somewhat sparse. The majority of studies have been carried out using dielectric barrier discharge plasma, and very few papers describe the use of any other type of plasma. This means that there is still a lot to be discovered in this area, which is novel and innovative.

Finally, it is worth mentioning that to the best of the authors’ knowledge, microwave discharge has not been tested as an ammonia decomposition method. Thus, the question arises: why not try to apply microwave plasma to the process, especially since microwave treatment is commonly used in thermal reactors? So, it could also potentially be used for the plasma treatment of ammonia. A characteristic feature of this type of plasma is the high efficiency of the vibrational excitation of the particles. This is especially important as it increases the energy efficiency of the plasma process. In combination with catalysts, this makes microwave plasma a promising tool for use in many industries.

## Figures and Tables

**Figure 1 ijms-24-14397-f001:**
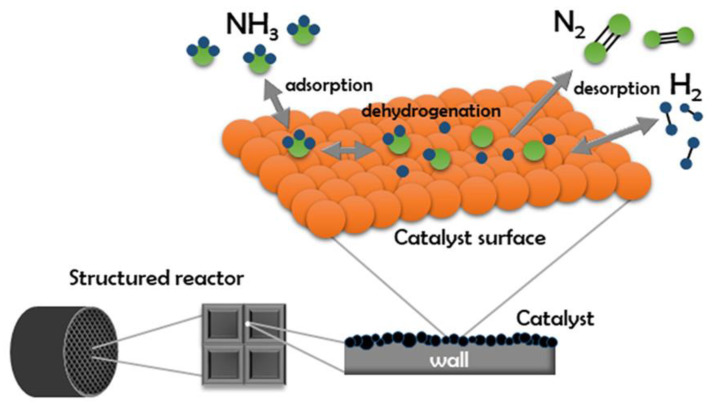
Decomposition on the catalyst surface. Used with permission from [25] (Creative Commons license).

**Figure 2 ijms-24-14397-f002:**
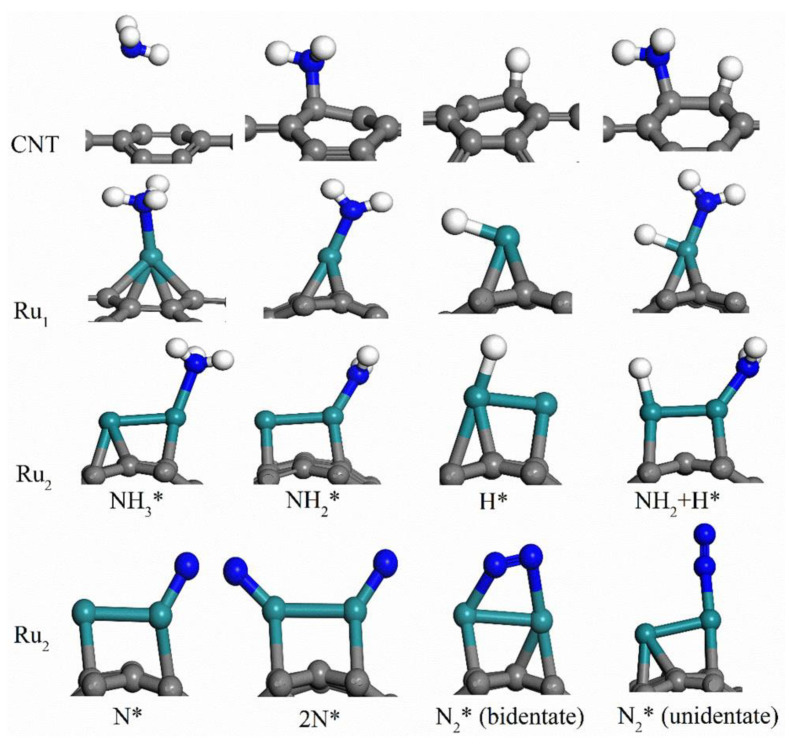
Adsorption geometry of pertinent species on CNT, Ru_1_@CNT, and Ru_2_@CNT. The C, Ru, N, and H atoms are colored gray, cyan, blue, and white, respectively. Used with permission from [30] (Copyright © 2018, American Chemical Society).

**Figure 3 ijms-24-14397-f003:**
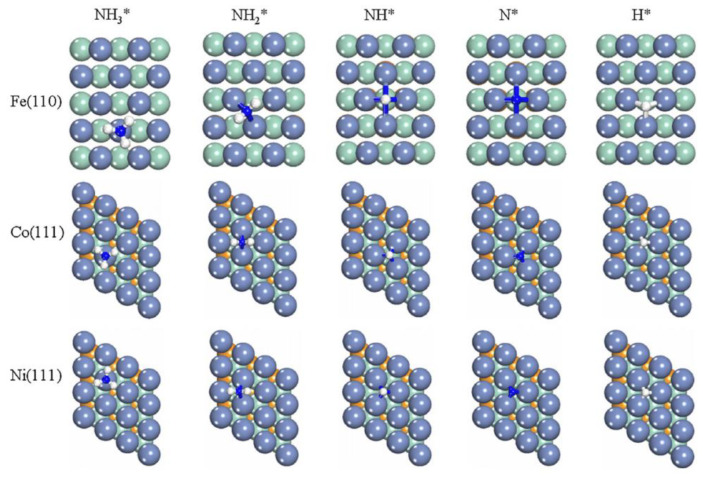
The most stable adsorption configurations of the surface of NHx (x = 0–3) and H on Fe(1 1 0), Co(1 1 1), and Ni(1 1 1). Used with permission from [32] (Copyright © 2012 Elsevier).

**Figure 4 ijms-24-14397-f004:**
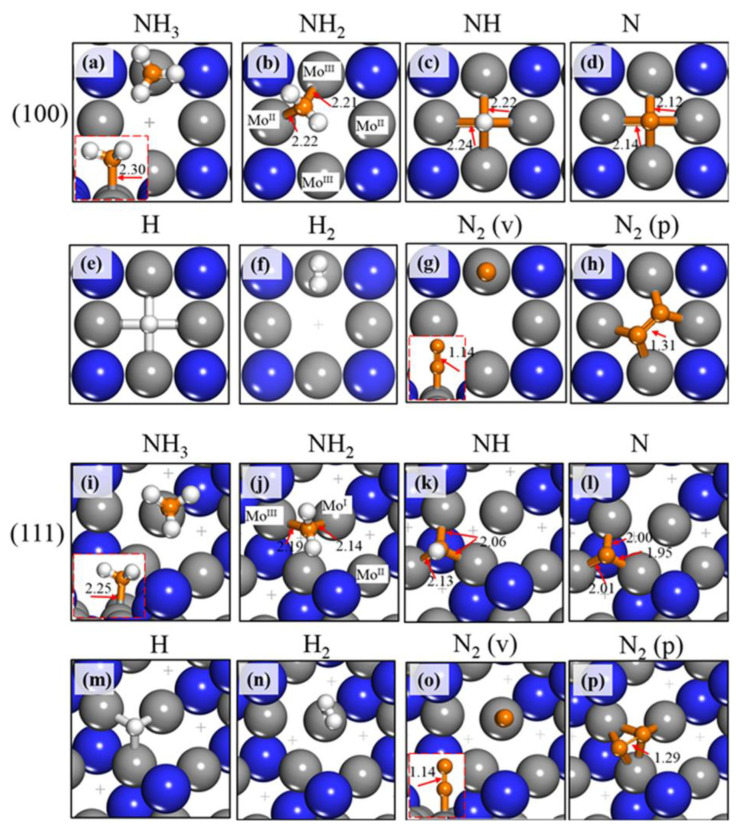
Most stable adsorption configurations of the NH_3_, NH_2_, NH, N, H, H_2,_ and N_2_ species on Mo_2_N(100) (**a**–**h**) and Mo_2_N (111) (**i**–**p**). The insets provide a side view of the corresponding adsorption configurations. “v” and “p” mean the vertical and parallel N_2_ adsorption configurations. The lengths of (Å) of the Mo–NH_x_ bonds and N–N bonds are given. Gray = Mo; blue = N; orange = N from NH_x_; white = H. Used with permission from [34] (Copyright © 2019, American Chemical Society).

**Figure 5 ijms-24-14397-f005:**
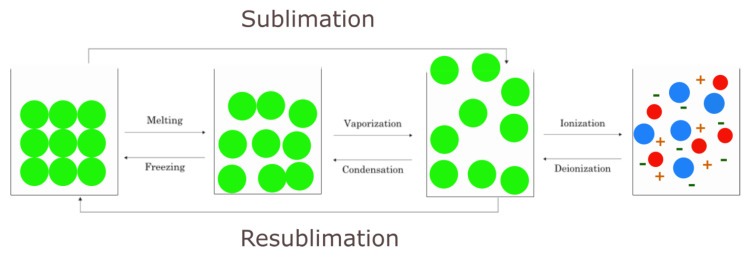
States of matter.

**Figure 6 ijms-24-14397-f006:**
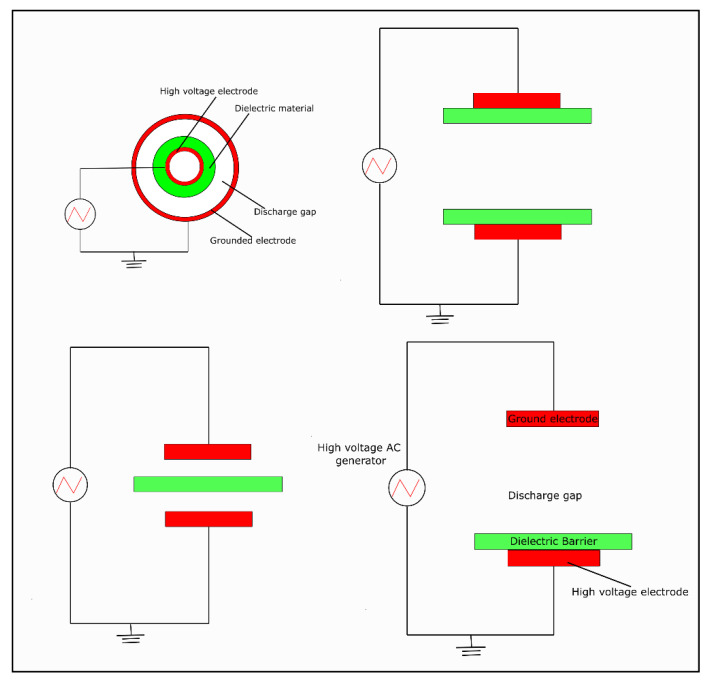
Common DBD reactor configurations.

**Figure 7 ijms-24-14397-f007:**
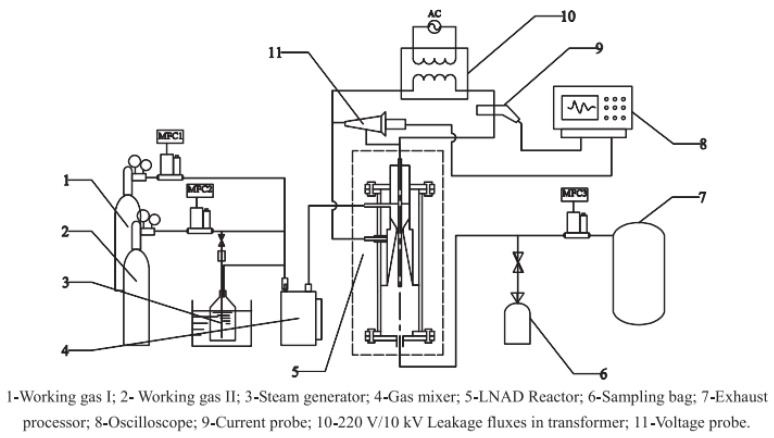
Diagram of the LNAD process. Used with permission from [83] (Copyright © 2014 Elsevier).

**Figure 8 ijms-24-14397-f008:**
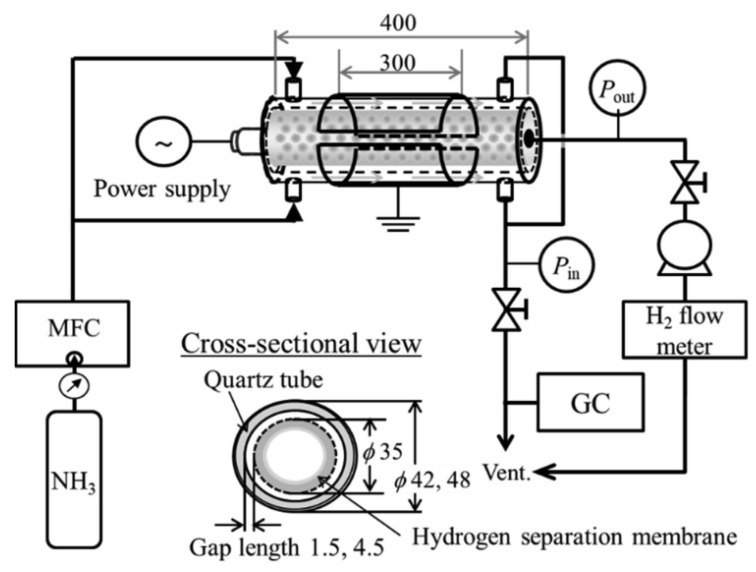
Experimental set-up for H_2_ separation and production using a plasma membrane reactor. Used with permission from [86] Copyright © 2019 Elsevier.

**Table 1 ijms-24-14397-t001:** Summary of hydrogen production pathways and colors.

	Grey	Blue	Turquoise	Green	Purple
Sources	Natural gas + Water, coal + Air	Natural gas + Water + Air	Natural gas + Water	Renewable energy + Water	Nuclear energy + Water
Process	SMR + coal gasification	SMR + CCS	Pyrolysis	Electrolysis	Electrolysis
Products	H_2_ + CO_2_	H_2_ + CO_2_ (partly captured)	H_2_ + C	H_2_ + O_2_	H_2_ + O_2_ + nuclear waste
Possibility of product separation	No	No	No	Yes	Yes

SMR—steam reforming of natural gas. CCS—carbon capture and storage.

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
