# Peer review of "Green Hydrogen Production through Ammonia Decomposition Using Non-Thermal Plasma"

_ijms, 2023, doi:10.3390/ijms241814397_

Round 1
Reviewer 1 Report
In this paper, the authors reviewed the ammonia decomposition systems focusing on non-thermal methods, especially under plasma conditions.
The paper is properly divided in sections and sub-sections, but it needs to be carefully revised and improved before being considered for publication in the journal.
- The authors should check the text since some errors are present;
- The authors should increase the literature survey by adding more recent papers.
- The statement at line 274, regarding the used of microwaves, needs a proper citation;
The authors should add more deep personal insights by giving a more critical summary of the reviewed literature;
The text contains some typos
Reviewer 2 Report
This review manuscript deals with the most recent achievements in the green hydrogen production by ammonia decomposition systems focusing on the non-thermal methods, especially under plasma conditions, but the catalytic methods are also shown. Since the new energy carriers (e.g. liuqid H2) play a significant role in transporting energy and have high impact on the environmentally friendly processes, this overview can be meaningful for the future developments on this field.
The manuscript is fairly written and the results are clearly presented. However, some minor revisions should be made before publication:
1) p. 8 line 229 The authors write that “Ruthenium is one of the noble metals, is relatively rare and very expensive.”, but this statement is ambiguous, because among the precious metal Ru is the cheapest one (see the attached chart), but it is more expensive than the base metals. Please, refine your affirmation.
2) p. 10 line 305 The authors also write that “This review focused on the low-temperature plasma process as a non-thermal method of ammonia decomposition to hydrogen and nitrogen.”, but it is not clear why you place this sentence here. Its logical order is questionable. Please, clarify this.
3) pp. 17–21 Style of the references does not meet the requirements of journal International Journal of Molecular Sciences. For example, in case of “Energy Conversion and Management: X 2022,15,100265.” you should use the following form: Energy Convers. Manage.: X 2022, 15, 100265. Practically, all references contain some mistakes. Please, check and modify them carefully.

The English also needs improvements. There are some, typical mistakes:
p. 1 line 29 „ … the environment's degradation.” instead of the degradation of environment
line 31 and elsewhere „… .[1]” instead of [1].
p. 2 line 67 „… on noble catalysts.” instead of on noble metal catalysts.
line 73 „… the electrode's potential.[8]” instead of the electrode potential [8].
line 77 „Contrarily, …” instead of On the contrary,
p. 4 line 147 „… the process's mechanism and kinetics.” instead of the mechanism and kinetics of process
p. 5 line 160 „… sites available - top, hollow, and bridge sites.” instead of sites available: top, hollow, and bridge ones.
p. 7 line 205 „It was proved recently …” instead of It has recently been proved
line 209 „… carbon materials' innate H-resistivity and N-affinity.” instead of innate H-resistivity and N-affinity of carbon materials.
p. 9 line 281 „Seyfeli et Al. …” instead of Seyfeli et al.
p. 16 line 567 „… NH3 …” instead of NH3
line 572 „… H2 and N2 …” instead of H2 and N2
Reviewer 3 Report
The manuscript submitted to IJMS entitled " Green hydrogen production through ammonia decomposition under non-thermal plasma" by Moszczyńska and co-workers presents a concise short review with an adequate number of references and information. The authors referenced and properly discussed the majority of relevant literature about this subject. This is a relevant topic with increasing significance in recent years, carefully prepared, concise, and a welcome addition to the literature. Therefore, this short revision has enough novelty/importance and should be considered for publication.
However, the conclusion would benefit if a personal point of view could be included. For instance, considering the most recent advancements, what are the possible future research directions?
Author Response
Dear Referee, thank you for the remark we corrected the Conclusions section
Reviewer 4 Report
The „1.2“ could be extended comparing ammonium could be extended as there are more technologies to store and transport hydrogen. In particular, liquid hydrogen carriers (for example toluene / methylcyclohexane, tibenzyltoluene, N-ethyl carbazole, etc.), methanol, synthetic methane, metal hydrides, etc.
The need for „2. Hydrogen fuel cells “and “2.2” were insufficiently justified and could be omitted.
Information in “4.2” (except info on membrane reactors) is partially provided in “3.2” and “3.3” and should be incorporated in “3.2-3.3” respectively or omitted.
The need for “6.4” is insufficiently justified. If there are no results on ammonium decomposition using microwave plasma maybe it is worth just mentioning it in conclusions as a possible new R&D direction?
A few general remarks: It remains insufficiently clarified the challenges are related to plasma-based methods for ammonium decomposition. Why there aren’t any of these methods on an industrial scale? Are there any discussions on the hydrogen price (CAPEX/OPEX) using the plasma approach in comparison to already used approaches (catalyst, high temperature based, etc.)?
Round 2
Reviewer 1 Report
The authors properly improved the manuscript.
All is ok
Author Response
Dear Referee, thank you
Reviewer 4 Report
The manuscript was properly improved following most of the reviewer's comments. However, the need for "2. Hydrogen fuel cells “and “2.1 - 2.2" remains insufficiently justified as H2 obtained from NH3 decomposition is free from oxygen and C-oxides. It is very well described and properly justified in "3. Ammonia decomposition mechanisms".
Author Response
Dear Referee,
In our opinion, chapter 2 - highlights the advantage of ammonia over other liquid H2 carriers. A reader unfamiliar with the topic will not notice this unique feature of NH3, which gives it an incredible advantage over such popular sources as CH4 or CH3OH. Hydrogen produced from the latter cannot be "green".
Moreover, by removal (ch.2) we lose the meaning of the title and the explanation of what the individual colors of hydrogen mean.
That is why I asked Editor to decide. If Editor decide so, with a heavy heart, but we will remove chapter 2.